# TOWARDS TEXT-GUIDED 3D SCENE COMPOSITION

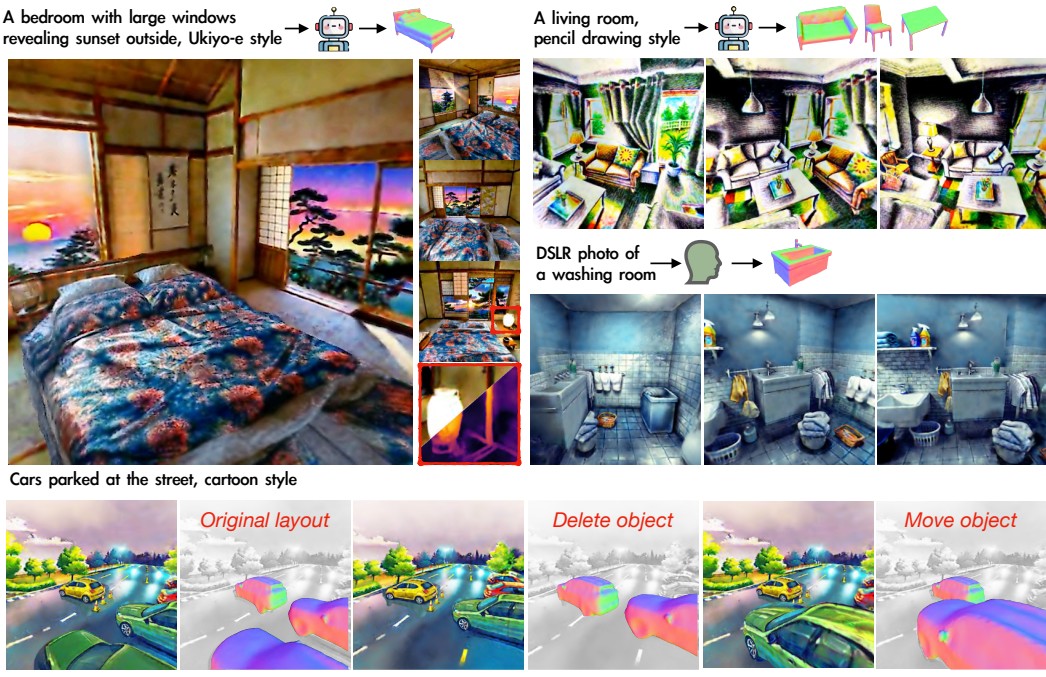

Figure 1: **Diverse 3D scenes synthesized by our method.** Our method can incorporate objects that are either automatically generated from text prompts (denoted as 🖥, top left, top right), or provided by a user (denoted as 🗣, middle right). Our method generalizes to different scene types and styles and supports scene manipulation, such as moving or deleting objects (bottom).

## ABSTRACT

We are witnessing significant breakthroughs in the technology for generating 3D objects from text. Existing approaches either leverage large text-to-image models to optimize a 3D representation or train 3D generators on object-centric datasets. Generating entire scenes, however, remains very challenging as a scene contains multiple 3D objects, diverse and scattered. In this work, we introduce SceneWiz3D – a novel approach to synthesize high-fidelity 3D scenes from text. We marry the locality of objects with globality of scenes by introducing a hybrid 3D representation – explicit for objects and implicit for scenes. Remarkably, an object, being represented explicitly, can be either generated from text using conventional text-to-3D approaches, or provided by users. To configure the layout of the scene and automatically place objects, we apply the Particle Swarm Optimization technique during the distillation process. Furthermore, in the text-to-scene scenario, it is difficult for certain parts of the scene (e.g., corners, occlusion) to receive multi-view supervision, leading to inferior geometry. To mitigate the lack of such supervision, we incorporate an RGBD panorama diffusion model, resulting in high-quality geometry. Extensive evaluation supports that our approach achieves superior quality over previous approaches, enabling the generation of detailed and view-consistent 3D scenes.

## 1 INTRODUCTION

Remarkable progress in text-to-3D (Poole et al., 2023; Wang et al., 2023a;c; Metzer et al., 2023; Chen et al., 2023; Lin et al., 2023a; Zhu & Zhuang, 2023) has been achieved in the generation of diverse and high-quality 3D objects, driven by techniques distilling knowledge from 2D foundation models (Rombach et al., 2022). In this work, we aim to go beyond the creation of individual 3D objects and embark on the synthesis of entire 3D scenes. Unlike a single object, a scene encapsulates a wealth of information, offering the potential for a truly immersive experience, especially vital for applications such as augmented reality (AR), virtual reality (VR), and filmmaking.

Generating 3D scenes entails dealing with numerous objects that have distinct appearances and are arranged in various layouts. Current text-to-3D methods often struggle to address these complexities, primarily due to unsuitable representations. Implicit representations, like Neural Radiance Field (NeRF) (Mildenhall et al., 2020), excel in modeling arbitrary scenes, yet face challenges in manipulating individual objects in scenes. In contrast, explicit representations, like meshes, provide explicit geometry and prove to be effective in representing objects, but are costly to maintain and update complex scenes with numerous objects. Taking advantage of both ends, we adopt a hybrid 3D representation with both implicit and explicit components. We employ Deep Marching Tetrahedra (DMTet) (Shen et al., 2021), an explicit 3D representation, for objects of interest, ensuring excellent multi-view consistency. Given text description, objects of interest can either be suggested by Large Language Models (LLM) or user-specified. Objects can then be initialized via off-the-shelf text-to-3D models. To represent the remaining scene elements, we utilize an implicit radiance field that provides flexibility in representing scenes with varying depth ranges.

Beyond 3D scene representations, obtaining proper 3D layouts, formatted as object configurations (i.e., positions, rotations, and scaling), remains a non-trivial task. Some concurrent efforts (Po & Wetzstein, 2023; Cohen-Bar et al., 2023) circumvent the challenge by taking predefined 3D layouts as inputs, which are less user-friendly compared to 2D layouts and underutilize the generalizability and creativity of foundation models. In contrast, we incorporate automatic updates of object configurations in the optimization process. The naive incorporation of update through back-propagation often falls into local minima, particularly for low-dimensional object configuration vectors. To circumvent this issue, we propose the use of Particle Swarm Optimization (PSO) (Kennedy & Eberhart, 1995) based on CLIP similarity. PSO is a particle-based optimization method inspired by the simulation of the movement and intelligence of swarms. It can effectively navigate the landscape of configurations through collaboration between different particles, evading local minima, and striking a good balance between exploration and exploitation.

Another challenge in 3D scene generation lies in obtaining detailed and complex geometry. Conventional distillation-based text-to-3D object generation approaches rely on 360-degree cameras placed around objects to provide multi-view information (Poole et al., 2023; Lin et al., 2023a). However, this approach is struggling with scene generation due to (1) camera placement within the scene, limiting viewing angles, especially for areas near scene boundaries, and (2) occlusions within scenes, restricting the coverage of viewing regions. Moreover, the longstanding Janus problem (i.e., the multi-face issue), originally identified in distillation-based 3D object generation (Hong et al., 2023), persists in scene generation, as shown in Fig. 6. To address these, we incorporate LDM3D (Stan et al., 2023), a diffusion model finetuned on panoramic images in RGBD space. During the optimization process, LDM3D provides additional prior information: The RGBD knowledge yields supervision in depth, while panoramic knowledge mitigates the issues of limited views with perspective images and disambiguates the global structure of a scene.

We dub our pipeline as SceneWiz3D. We thoroughly evaluate the proposed method in terms of appearance and geometry by employing the CLIP similarity metric (Radford et al., 2021b), a depth estimator (Ranftl et al., 2022) and the FID metric (Heusel et al., 2017) on both perspective and panoramic views. SceneWiz3D achieves state-of-the-art performance in text-to-3D scene generation compared to all baseline methods. SceneWiz3D effectively synthesizes scenes based on a wide range of user-provided text prompts, while also accommodating specific user preferences for 3D assets and seamlessly arranging the selected objects within the scenes.

## 2 RELATED WORK

**Text-to-3D object generation.** It is intuitive to learn a text-to-3D generative model using 3D data. Recent research has explored various 3D representations, such as point clouds (Nichol et al., 2022), signed distance functions (Cheng et al., 2023), triplane representations (Wang et al., 2023b), and neural representations (Jun & Nichol, 2023), etc. Despite these advancements, current text-to-3D methods still face limitations in terms of quality and diversity, primarily due to a lack of annotated data. In contrast, 2D generative models have made significant improvements in visual quality and diversity, owing to diffusion models (Ho et al., 2020; Nichol & Dhariwal, 2021; Dhariwal & Nichol, 2021; Rombach et al., 2022) trained on large-scale annotated 2D image datasets (Schuhmann et al., 2022). Therefore, it is crucial to bridge the gap between 2D and 3D generative models, and several strategies have emerged to achieve this. Score Distillation Sampling (SDS) (Lin et al., 2023a; Wang et al., 2023a; Chen et al., 2023; Wang et al., 2023c; Zhu & Zhuang, 2023; Wang et al., 2023a), introduced by Dreamfusion (Poole et al., 2023), employs pretrained text-to-image diffusion models as priors for optimizing parametric spaces. Lastly, Zero-1-to-3 (Liu et al., 2023b) finetunes the Stable Diffusion model (Rombach et al., 2022) with 3D data to gain control over camera perspective, enabling novel-view synthesis and 3D reconstruction (Liu et al., 2023a; Qian et al., 2023).

**3D scene generation.** Generating 3D scene models presents a unique challenge, as collecting 3D scene data is inherently difficult compared to 2D images, whether through scanning or manual creation. Inspired by Neural Radiance Fields (NeRF) (Mildenhall et al., 2020), NeRF-based generators (Chan et al., 2022; 2021; Gu et al., 2022; Schwarz et al., 2020; Siarohin et al., 2023; Skorokhodov et al., 2023; Xu et al., 2023) in conjunction with GAN-based framework (Goodfellow et al., 2014; Karras et al., 2020) emerged as dominant approaches to learning 3D scene structure from 2D image collections. Among these models, GSN (DeVries et al., 2021) was an early attempt at scene-level synthesis by modeling traversable indoor scenes using local radiance fields, and InfiniCity (Lin et al., 2023b) proposed a pipeline consisting of 2D and 3D models to make use of both types of data. Recent advancements, facilitated by diffusion models, have introduced strategies to lift 2D knowledge into 3D representations, achieving 3D scene synthesis. The lifting from 2D images to 3D scenes can occur either through jointly predicting depth information (Stan et al., 2023) or by utilizing off-the-shelf methods for depth estimation (Höllein et al., 2023). Some concurrent efforts (Po & Wetzstein, 2023; Cohen-Bar et al., 2023) take predefined 3D layouts as inputs to sidestep the difficulties of compositions in 3D scene generation. All these methods encounter challenges related to inaccurate scene-level geometry.

## 3 METHOD

Our goal is to create high-fidelity 3D scenes from text. We build our method based on existing text-to-3D object approaches with score distillation sampling (Sec. 3.1). Different from individual object which is local and compact, scenes can be global with objects scattered throughout, which poses a challenge for a unified 3D representation for generation. Thus, we marry the locality of objects with the globality of scenes by introducing a hybrid 3D representation (Sec. 3.2). , We discuss how to optimize the hybrid representation in Sec. 3.3. To automatically configure the layout of the scene, we use Particle Swarm Optimization, presented in Sec. 3.3.1. To mitigate the lack of multi-view supervision caused by occlusion, we incorporate a pre-trained RGBD panorama diffusion model for providing additional guidance, illustrated in Sec. 3.3.2.

### 3.1 PRELIMINARIES

Recent advancements in text-to-3D object synthesis have demonstrated the effectiveness of distilling knowledge from large text-to-image diffusion models (Rombach et al., 2022; Saharia et al., 2022). A text-to-image diffusion model contains a denoising autoencoder $\epsilon_\phi$, parameterized by $\phi$. A noisy data $\mathbf{x}_t$ is created by adding sampled noise $\epsilon \in \mathcal{N}(\mathbf{0}, \mathbf{I})$ over the original image $\mathbf{x}$. Given $\mathbf{x}_t$, time step $t$, and text embedding $\mathbf{y}$, the autoencoder estimates the sampled noise $\epsilon$, denoted as $\hat{\epsilon}_t = \epsilon_\phi(\mathbf{x}, \mathbf{y}, t)$. We will omit the time step $t$ for simplicity. A weighted denoising score matching objective (Ho et al., 2020; Kingma et al., 2021) is derived as the training objective of the diffusion model: $\mathcal{L}_{\text{diff}}(\phi, \mathbf{x}) = \mathbb{E}_{t, \epsilon}[w(t) \| \hat{\epsilon} - \epsilon \|_2^2]$, where $w(t)$ is a weighting term. In text-to-3D synthesis, Score Distillation Sampling (SDS) (Poole et al., 2023) is used to optimize a 3D representation which is parameterized by $\theta$, using as gradients the denoising score on rendered images $\mathbf{x}$, denoted as

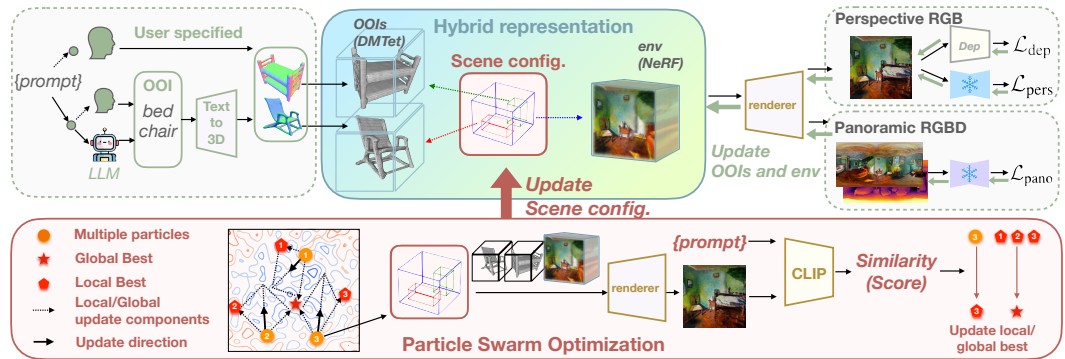

Figure 2: **SceneWiz3D Overview.** To model 3D scenes, we adopt a hybrid representation containing explicit and implicit components: DMTets for objects of interest (OOIs) and NeRF for the environment. Given a text prompt, we first identify OOIs of the scene, and initialize their DMTets. We update the OOIs' configurations with Particle Swarm Optimization based on CLIP similarity, and update both OOIs and the environment by score distillation with a text-to-image diffusion model, a panoramic RGBD diffusion model, along with a depth regularizer.

$\mathbf{x} = g(\theta)$. Formally, the gradient is denoted as:

$$\nabla_\theta \mathcal{L}_{\text{SDS}}(\phi, \mathbf{x} = g(\theta)) = \mathbb{E}_{t \sim \mathcal{U}(0,1), \epsilon \sim \mathcal{N}(\mathbf{0}, \mathbf{I})} \left[ w(t)(\hat{\epsilon} - \epsilon) \frac{\partial \mathbf{x}}{\partial \theta} \right]. \tag{1}$$

Wang et al. (2023c) propose to model the 3D parameter as a random variable instead of a constant as in SDS and present variational score distillation (VSD):

$$\nabla_\theta \mathcal{L}_{\text{VSD}}(\phi, \mathbf{x} = g(\theta)) = \mathbb{E}_{t \sim \mathcal{U}(0,1), \epsilon \sim \mathcal{N}(\mathbf{0}, \mathbf{I})} \left[ w(t)(\hat{\epsilon} - \epsilon_{\text{LoRA}}(\mathbf{x}, \mathbf{y}, t)) \frac{\partial \mathbf{x}}{\partial \theta} \right], \tag{2}$$

where $\epsilon_{\text{LoRA}}$ is the low-rank adaptation of the pretrained diffusion model. VSD can help alleviate the over-saturation, over-smoothing, and low-diversity problems met in SDS.

## 3.2 HYBRID SCENE REPRESENTATION

A scene typically consists of many objects arranged in various layouts. Different scenes also have varying depth ranges. As a result, it is challenging to find a unified representation to model an entire 3D scene effectively. For example, implicit representations like NeRF can result in aliasing and floating artifacts and fail to model crisp object surface. While explicit representations, like meshes, are costly to maintain and update when representing a whole scene with multiple objects. To address this, we propose to employ a hybrid scene representation with both implicit and explicit components to handle the complexity of a single scene, which is detailed below.

**DMTet for objects of interest.** We identify objects of interest (OOIs) for target scenes and model those objects with the explicit representation, DMTet. Specifically, DMTet maintains an explicit surface via deep marching over the signed distance value predicted on vertexes. Such explicit representation ensures OOIs maintaining multi-view consistency. As OOIs often occupy significant regions of the rendered image, maintaining their consistency greatly contributes to the overall scene fidelity. Moreover, the categories of OOIs can either directly identified by users, or automatically determined with the help of the Large Language Model (LLM) with queries like "*Please suggest several ordinary objects in a scene. The scene can be described as {prompt}.*" With the recommended categories of OOIs, such as beds, we can use text-to-3D object generative models to initialize the generation of these objects for the DMTet representation.

To effectively model different objects of interest, we use separate DMTet for each object. Each DMTet contains two networks $\Phi_g$ and $\Phi_c$ which take hashing-based encoding as input to model the geometry and color separately. Specifically, the geometry network $\Phi_g$ predicts the signed distance function (SDF) value for each vertex. We initialize the network by optimizing it to fit the SDF of the instantiated 3D objects. We use Marching Tetrahedra to extract the triangular mesh from this representation. Then we can get the rendered occupancy mask, normal map, and depth map by

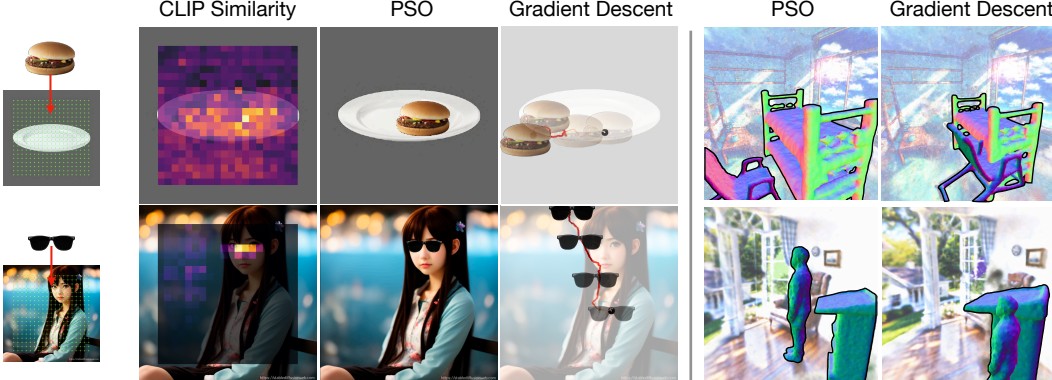

Figure 3: **Discovering scene configurations.** Naive gradient-based optimization suffers from local minima imposed by low-dimensional and non-convex optimization space, leading to improbable configurations for both 2D and 3D (objects overlap or are incorrectly placed). Particle Swarm Optimization, instead, correctly identifies a plausible configuration for both 2D and 3D.

differentiable rasterization. Finally, the color network $\Phi_c$ predicts the color of each point in the occupancy mask, which forms the RGB observation $I_{\text{fg}}$.

**NeRF for the environment.** We model the remaining scene components, termed as environment, with an implicit representation, NeRF. NeRF offers versatility in handling scenes with complex layouts and varying depth ranges. It can effectively accommodate both bounded and unbounded scenes (Zhang et al., 2020; Barron et al., 2022; 2023), and is flexible enough to render complex lighting effects that is otherwise not trial to implement with explicit representation. We model the environment as a volumetric radiance field that predicts density and color for any given 3D point. We follow NeRF's volumetric rendering equation to integrate density and colors along each ray, resulting in the rendered image, $I_{\text{bg}}$, contributed by the environment.

After separately predicting the geometry and color of the foreground OOIs and the background environment, we use z-buffering to blend the rendered results together to form the final image.

## 3.3 GENERATING SCENES

Our method requires only a text prompt to generate a scene. We first determine which 3D object categories need to be generated by querying an LLM (Touvron et al., 2023; Brown et al., 2020), where a text-to-image diffusion model (Rombach et al., 2022) can be used to generate images of suggested objects. After that, an off-the-shelf image-to-3D model (Cheng et al., 2023) generates 3D objects of interest. We note that the proposed process is flexible, accommodating alternative text-to-3D pipelines and allowing users to incorporate their preferred 3D objects at any stage of the process, as shown in Fig. 1. After obtaining the 3D objects of interest, we then initialize our hybrid scene representation. We initialize the geometry of DMTets to match the 3D objects obtained, and initialize the environment NeRF model by assigning a predefined density distribution designed for scenes following ProlificDreamer (Wang et al., 2023c). At last, we generate the scene by alternating between optimizing the scene configuration (i.e., object poses, see Sec. 3.3.1) and optimizing the scene model (i.e., DMTets for OOIs and NeRF for the environment, see Sec. 3.3.2).

### 3.3.1 AUTOMATICALLY LEARNING SCENE CONFIGURATIONS

As we disentangle OOIs from the rest of the scene in our hybrid representation, we need to determine the configuration for each object, location, scaling, and rotation. However, due to the complex and diverse layouts of scenes, obtaining reasonable configurations poses a challenge. Previous works (Huang et al., 2023; Tang et al., 2023) have attempted to learn scene configurations or layouts from limited datasets (e.g., indoor scenes), limiting the generalizability. Instead, we propose leveraging multi-modal foundation models for enhanced generalization.

**Scene configuration score.** We assess a scene configuration with the CLIP (Radford et al., 2021a) similarity between rendered images generated by the configuration and the input text prompt. Since

our rendering pipeline is fully differentiable, a straightforward idea is to update the configuration with backpropagation. However, as we observed, such an approach is prone to getting trapped in a local minima, as scene configurations represent a very low-dimensional space for gradient-based optimization. We illustrate the finding in 2D layouts (Fig. 3, left) and 3D layouts (Fig. 3, right).

In our initial experiments, we also examined the use of SDS loss for optimizing scene configuration. We encountered similar optimization challenges as described earlier. Furthermore, we observed that the global minimal point of the SDS score map does not correspond to the optimal configuration. This discrepancy arises from the fact that SDS is a proxy loss used to estimate the gradient of the KL divergence objective (Poole et al., 2023), and itself does not effectively distinguish between proper and improper configurations.

**Particle swarm optimization.** To overcome this challenge, we employ a non-gradient-based heuristic searching approach called Particle Swarm Optimization (PSO). PSO is a particle-based optimization algorithm that simulates the social behavior of a swarm of particles. During the $n$-th iteration, each particle $\mathbf{a}_i(n)$ represents a potential scene configuration. Its velocity $\mathbf{v}_i(n)$ determines its movement in the search space. We update each particle as:

$$\mathbf{v}_i(n+1) = k \cdot \mathbf{v}_i(n) + c_1 \cdot r_1 \cdot (\mathbf{pbest}_i - \mathbf{a}_i(n)) + c_2 \cdot r_2 \cdot (\mathbf{gbest} - \mathbf{a}_i(n)), \quad (3)$$

$$\mathbf{a}_i(n+1) = \mathbf{a}_i(n) + \mathbf{v}_i(n+1) \quad (4)$$

where $\mathbf{pbest}_i$ is the best position found by the particle, $\mathbf{gbest}$ is the best position found by any particle in the swarm, $k, c_1, c_2$ are hyper-parameters, and $r_1, r_2$ are random numbers used for exploration. PSO can effectively explore the scene configurations' search space and converge towards an optimal solution. We disable gradient descent over scene configurations and perform PSO updates every 3000 iterations.

### 3.3.2 OPTIMIZING SCENE MODEL

To optimize the parameters of a scene, we render perspective RGB image and panoramic RGBD images. Then, we employ VSD guidance with Stable Diffusion on perspective images, and SDS guidance with LDM3D on panoramic images, denoted as $\mathcal{L}_{\text{pers}}$ and $\mathcal{L}_{\text{pano}}$, respectively. Additionally, we include a depth regularization loss to further improve the quality of geometry.

The overall optimization target is a weighted sum of the above loss terms:

$$\mathcal{L} = \lambda_{\text{pers}}\mathcal{L}_{\text{pers}} + \lambda_{\text{pano}}\mathcal{L}_{\text{pano}} + \lambda_{\text{dep}}\mathcal{L}_{\text{dep}}, \quad (5)$$

where $\lambda_{\text{pers}}$, $\lambda_{\text{pano}}$, and $\lambda_{\text{dep}}$ are weighting coefficients. Below, we discuss details of $\mathcal{L}_{\text{pano}}$, $\mathcal{L}_{\text{dep}}$, and refer readers to ProlificDreamer (Wang et al., 2023c) for details of applying VSD for $\mathcal{L}_{\text{pers}}$.

**Score distillation sampling in RGBD panoramic view ($\mathcal{L}_{\text{pano}}$).** Multi-view supervision plays a crucial role in accurately reconstruct 3D structure. In the context of text-to-3D scene generation, viewing angles are constrained as cameras are positioned within scenes, leading to frequent occlusions that prevent certain regions from receiving comprehensive multi-view supervision. Consequently, this can result in foggy and distorted generated geometry. To address this deficiency, we propose to directly leverage supervision from a geometry-aware diffusion model, LDM3D (Stan et al., 2023). LDM3D is specifically designed to denoise data in RGBD space. By rendering RGBD images and distilling knowledge from LDM3D in a similar manner as in Score Distillation Sampling with Stable Diffusion, we can enhance the geometry of the synthesized scene. In addition, a version of LDM3D is finetuned over panoramic equirectangular images, able to generate realistic panoramas representing different scene types, providing holistic information about scene structure. To adopt panoramic views into the SDS pipeline, we need to modify the casting rays during rendering. The ray direction cast from the pixel $(u, v)$ is computed by converting its spherical coordinates to Cartesian coordinates. The vertical and horizontal viewing angles can be defined as $\pi v/H, 2\pi u/W$, where $H, W$ are the image resolution. Note that since we aim to improve the geometry, and OOIs modeled by DMTet already enjoy a good geometry guarantee, we do not render OOIs in the panorama view.

**Depth regularization ($\mathcal{L}_{\text{dep}}$).** We also adopt a depth regularizer $\mathcal{L}_{\text{dep}}$ based on a monocular depth estimator, MiDaS (Ranftl et al., 2022), to further improve the geometry property of the environment NeRF. Specifically, in perspective view, we render RGB image $I$ and disparity image $I_d$ by NeRF rendering. We use the MiDaS to predict a disparity image $\hat{I}_d$ based on $I$. The MiDaS network is frozen during optimization. The regularization term $\mathcal{L}_{\text{dep}}$ is designed as the least square distance between $I_d$ and $\hat{I}_d$, formally written as $\mathcal{L}_{\text{dep}} := \min_{s,b \in \mathbb{R}} \|sI_d + b - \hat{I}_d\|_2^2$.

Table 1: **Quantitative evaluation.** We evaluate the proposed methods against (a) the baseline methods and (b) different design choices in terms of the appearance alignment between outputs and prompts, and in terms of geometry using an off-the-shelve depth estimator and FID score.

| | Appearance | Geometry | |
|---|---|---|---|
| | CLIP-AP↑ | align↓ | FID↓ |
| DreamFusion | 92.6 | 0.25 | 41.18 |
| ProlificDreamer | 95.5 | 0.30 | 76.22 |
| Text2room | 63.4 | 0.32 | 26.25 |
| LDM3D | 69.3 | 0.24 | 11.33 |
| Ours | **97.6** | **0.14** | **10.43** |

(a) **Comparison with baselines.**

| | $\mathcal{L}_{pers}$ | $\mathcal{L}_{pano}$ | $\mathcal{L}_{dep}$ | PSO | Appearance CLIP-AP↑ | Geometry align↓ | FID↓ |
|---|---|---|---|---|---|---|---|
| | | ✓ | | | 59.7 | 0.23 | 23.89 |
| SDS | | | | ✓ | 90.5 | 0.26 | 38.93 |
| SDS | ✓ | | | ✓ | 96.7 | 0.22 | 29.41 |
| VSD | | | | ✓ | 93.0 | 0.32 | 68.53 |
| VSD | ✓ | | | ✓ | 96.1 | 0.24 | 28.39 |
| VSD | ✓ | ✓ | | | 91.8 | 0.19 | 18.92 |
| VSD | ✓ | ✓ | | ✓ | **97.6** | **0.14** | **10.43** |

(b) **Ablation study.**

## 4 EXPERIMENTS

### 4.1 SETTINGS

**Implementation details.** The geometry and color networks in NeRF and DMTet are parameterized as multi-layer perceptrons (MLPs) with hashing-based positional encoding (Müller et al., 2022). For score distillation, we use a pretrained Stable Diffusion model[1] for perspective views and LDM3D-pano for panoramic views[2], and adapt the time step sampler from DreamFusion (Poole et al., 2023). Optimization comprises 20,000 iterations with the Adam optimizer, conducted on a single A100 GPU with a batch size of 1. Further implementation details are provided in the Supplementary.

**Test set.** To compare with other text-to-3D scene synthesis approaches, we generate a set of 10 test prompts describing various indoor scenes, such as museums and classrooms (see supplementary for the full list). For each scene, we sample 1000 different camera viewpoints, yielding a total of 1000 images. Different from previous works (Höllein et al., 2023), our method is flexible to generate versatile scenes. For ease of comparison, we choose indoor scenes only.

**Metrics.** We assess the quality of text-to-3D scene synthesis from two different perspectives. **(1) appearance:** We evaluate the quality of the rendered images by measuring their alignment with the provided captions, using an average precision of the CLIP similarity between the rendered images and their co captions. **(2) geometry:** Assessing the quality of generated geometry is non-trivial. Following EG3D (Chan et al., 2022), we measure the alignment between the rendered perspective-view disparity map, denoted as $I_d$, and the disparity map $\hat{I}_d$ predicted by MiDaS (Birkl et al., 2023), using the corresponding rendered RGB image as input. We quantify the alignment using mean square error (MSE). As both disparity maps $I_d$, $\hat{I}_d$ are non-metric, we adjust the rendered disparity map to align with MiDaS's by solving a least square problem i.e., $\min_{s\in\mathbb{R}^+,b\in\mathbb{R}} \|sI_d + b - \hat{I}_d\|_2^2$, where $s$, $b$ are parameters for alignment. Additionally, we calculate Fréchet Inception Distance (FID) (Heusel et al., 2017) to assess the visual quality of the rendered perspective-view disparity map. We utilize the NYU-dep-v2 (Nathan Silberman & Fergus, 2012) as the ground truth source for calculating the FID score. We find that the FID score is sufficiently sensitive to artifacts resulting from the presence of floating densities and opaque edges.

**Baselines.** We compare our method with DreamFusion (Poole et al., 2023) and Prolific-Dreamer (Wang et al., 2023c), two SDS-based methods originally designed for text-to-3D object synthesis. ProlificDreamer extends its application to text-to-3D scene synthesis by introducing a density bias and a different camera sampling strategy. We apply the same setting to DreamFusion. In addition to these SDS-based methods, we also compare our method with two approaches that do not rely on SDS for creating 3D scenes. Specifically, in Text2room (Höllein et al., 2023), a text-conditioned inpainting model combining with monocular depth estimation is used to iteratively create and refine a textured mesh. Moreover, we compare with LDM3D (Stan et al., 2023) which we leverage as additional guidance. We follow the steps specified in the original paper to convert its generated RGBD panorama image into a 3D mesh using TouchDesigner (Derivative, 2023) for perspective view synthesis.

---

[1]https://huggingface.co/stabilityai/stable-diffusion-2-1
[2]https://huggingface.co/Intel/ldm3d-pano

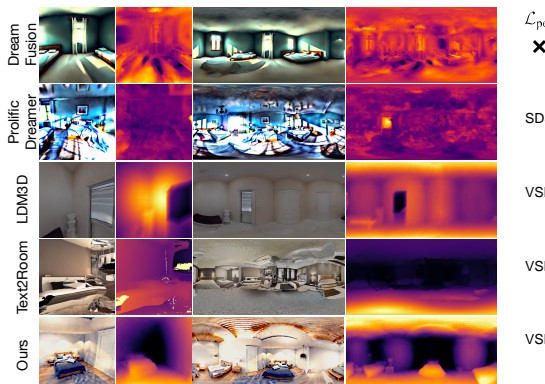 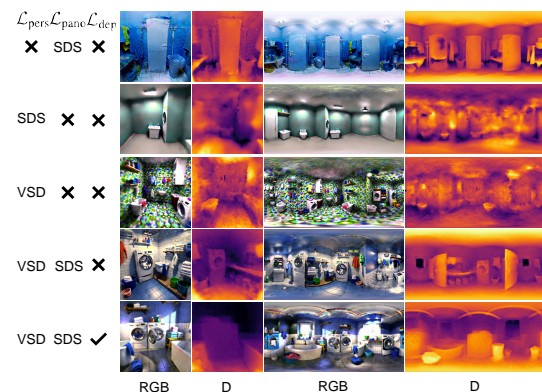

Figure 4: **Qualitative baselines comparisons.** prompt: *a bedroom, realistic detailed photo*.

Figure 5: **Qualitative ablation study.** prompt: *a washing room, realistic detailed photo*.

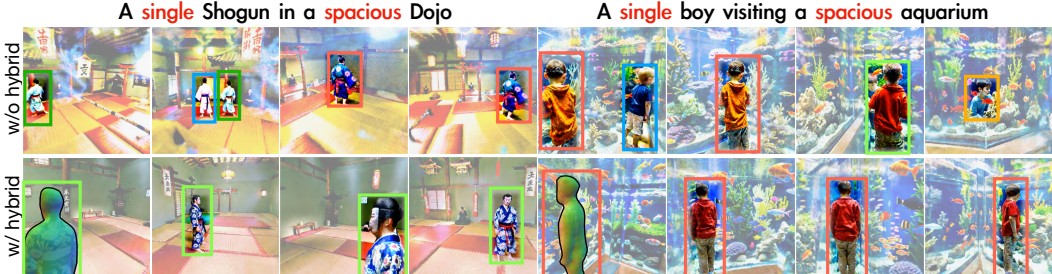

Figure 6: **Janus problem** in scene synthesis is expressed as multiple instances of the same object category (top row), which can be mitigated with hybrid representation (bottom row). Each bounding box color represents a single identity.

## 4.2 MAIN RESULTS

As reported in Tab. 1(a), our method consistently outperforms existing methods in terms of appearance and geometry quality. We provide a qualitative comparison in Fig. 4 by visualizing synthesized RGB and disparity images from both perspective and panoramic views. Due to inadequate multi-view supervision, the disparity maps synthesized by DreamFusion and ProlificDreamer contain fuzziness, opaqueness, and floating artifacts. Furthermore, DreamFusion's output lacks fine-grained details. As LDM3D and Text2room both create 3D environments by projection from 2D, they suffer from severe distortion issues. In contrast, our proposed method successfully synthesizes scenes with exceptional details. The disparity maps obtained by our method exhibit greater realism, signifying a more plausible geometry for the synthesized scene.

## 4.3 ABLATION STUDY

In Tab. 1(b), we systematically perform ablation experiments for our approach.

**PSO for discovering scene configurations.** In the final two rows of Tab. 1(b), there is a significant drop in CLIP-AP when substituting PSO with gradient descent. The score decreases from 97.6 to 91.8, demonstrating the essential role of PSO in searching reasonable configurations in the low dimensional configuration space.

**Perspective view guidance.** We experiment with synthesizing a scene only with panoramic view SDS guidance, and observe a decrease in appearance quality (first row of Tab. 1(b) and Fig. 5). By incorporating perspective view guidance, the appearance can be enhanced. This enhancement improves the similarity between rendered views and the prompts, and leads to a CLIP-AP score that exceeds 90. Additionally, we find that SDS tends to synthesize over-simplified scenes. This simplicity can make it easier to predict the depth and subsequently leads to a marginal performance advantage for SDS over VSD. Nevertheless, we still opt for VSD as our preferred guidance for the perspective view due to its ability to capture and incorporate greater levels of detail.

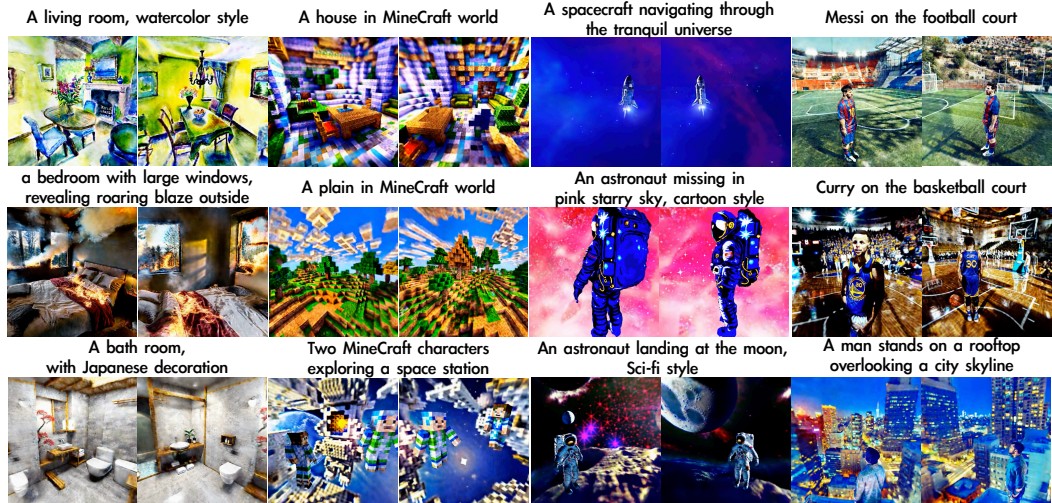

Figure 7: **Diverse scenes generated by `SceneWiz3D`.** Our method successfully generalizes to different scene types (indoor, outdoor) with varying styles (realistic, cartoon, painting) and OOI categories (human, object).

**Panoramic RGBD guidance.** We observe that the introduction of panoramic view guidance reduces floating artifacts and increases the precision of the geometry, consequently benefiting the appearance quality. This improvement can be attributed to the denoising capability within the RGBD space, offering valuable geometry-aware feedback.

**Depth regularizer.** The depth regularizer serves as a direct supervision mechanism by measuring the distance between the rendered and MiDaS predicted disparity map. Leveraging the robust capabilities of MiDaS, we find that this term enhances the quality of geometry, reducing the disparity map's alignment and FID score by a large margin (see last row of Tab. 1(b)).

### 4.4 Further Analysis

**Alleviating the multi-faces problem.** We find that the multi-faces (Janus) problem, originally discovered in text-to-3D object scenario, also exists in the scene scenario. As shown in Fig. 6, when given the prompt: *a single boy visiting a spacious aquarium*, multiple instances of boys appear throughout the scene. However, by explicitly modeling objects of interest, such as the boy in this case, we can mitigate the issue of each perspective view generating new objects. Consequently, the Janus Problem is alleviated.

**Generalizability to different scene types.** We observe that our method generalizes well across various scene types, as shown in Fig. 7.

**Limitations.** Our method shares common limitations with other SDS-based approaches, including long optimization time and color saturation. Despite significant improvements, our approach may still occasionally result in foggy or distorted geometry. Moreover, our current scene configuration optimization is limited by the capabilities of the CLIP model, which does not excel in fine-grained manipulation tasks. Finally, our approach is restricted by the capabilities of the LDM3D model, which currently lacks support for fine-grained context generation and artistic styles. We anticipate that more advanced RGBD and panoramic diffusion models could further enhance the quality of our method.

### 5 Conclusion

We present a novel pipeline, `SceneWiz3D`, for text-to-3D scene generation, utilizing a hybrid of implicit and explicit representation which generalizes well across diverse scene types. We demonstrate the effectiveness of PSO in optimizing scene configurations, and the incorporation of score distillation from both perspective RGB and panoramic RGBD view notably enhances scene generation quality. `SceneWiz3D` achieves state-of-the-art 3D scene generation quality in terms of both appearance and geometry details.

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

APPENDIX

This appendix is organized as follows. Appendixes A and B present the implementation details of our proposed SceneWiz3D and baselines respectively. Appendix D justifies the usage of FID over disparity map to assess the geometry of synthesized scenes. Appendix C lists all the prompts used for evaluation. Appendix E demonstrates that SceneWiz3D is capable of synthesizing artistic scenes with diverse styles. We also show qualitative comparison and a collection of diverse synthesized scenes on the anonymous website: https://scenewiz3d.github.io.

## A    IMPLEMENTATION DETAILS OF SCENEWIZ3D

**Perspective guidance.** We use VSD as our perspective view guidance. We inherit the original camera sampling strategy and annealed time schedule for score distillation.

**Panoramic RGBD guidance.** Different from perspective camera that is sampled on a sphere and looks at the middle of the scene, our panoramic camera is placed at the center of the scene with a small random offset. The magnitude of this offset is limited to a maximum of ten percent of the scene radius. Since the panoramic guidance does not consider rendering Object of Interests (OOIs) in the panoramic view, it lacks awareness of the presence of OOIs. To avoid conflicts between the panoramic guidance and OOIs, we exclude the panoramic guidance from the initial 5000 iterations. After the first 5000 iterations, we introduce the panoramic guidance based on the rough scene layout obtained from the perspective guidance. To ensure a smooth transition, we gradually anneal the maximum time step from 0.5 at 5000 iterations to 0.3 at 20000 iterations. We render the panoramic image in $256 \times 512$ resolution. Furthermore, we also exclude the panoramic guidance during the last 5000 iterations. We have observed that incorporating it during this stage can slightly compromise the visual quality of the scene.

**Depth regularizer $\mathcal{L}_{\mathbf{dep}}$.** We use the official *dpt-beit-large-512* version of MiDaS to estimate the target depth for calculating the depth regularizer term.

**Coefficients.** $\lambda_{\text{pers}}$, $\lambda_{\text{pano}}$, and $\lambda_{\text{dep}}$ are set to $1, 10^{-1}, 10^4$ for all experiments.

## B    IMPLEMENTATION DETAILS OF BASELINES

**ProlificDreamer (Wang et al., 2023c).** We adopt the implementation from threestudio[3] which achieves a similar visual quality to the results in the original paper. We inherit the camera sampling scheme and density initialization as proposed in the original paper. We only render $64 \times 64$ resolution images for the first 5000 iterations, and then render in $512 \times 512$ images for another 20000 iterations.

**DreamFusion (Poole et al., 2023).** We implement DreamFusion based on the ProlificDreamer's implementation specified above. We preserve all the configs, except for the modification of the guidance term from VSD to SDS.

**Text2room (Höllein et al., 2023).** We train text2room using our text prompts, following its official guidance. We generated panoramic images and depth maps using Blender. As the mesh color is assigned to each vertex, we rendered the panoramic images without additional lighting.

**LDM3D (Stan et al., 2023).** We follow the official implementation of LDM3D. We first synthesize a panoramic RGBD image by LDM3D-pano. Then we use TouchDesigner to convert it into a 3D mesh for rendering novel-view images. Fig. A1 illustrates the process in details. The rendering pipeline processes the input depth map as a height map to deform a 3D sphere with a radius of 1, using the input image as the texture map for the sphere. The degree of deformation is controlled by the 'displacement scale' parameter in the Phong shader, which we empirically set to 1 to minimize distortion in perspective view rendering. We position the camera on a circle with a radius of 0.4 and ensure it always points towards the scene's center. Rendering RGB and depth images for the perspective view is straightforward using the renderer TOP. For panoramic views, we configure the renderer TOP to produce dual paraboloid images and then use a projection TOP to convert them into equirectangular panorama images.

---

[3]https://github.com/threestudio-project/threestudio#prolificdreamer

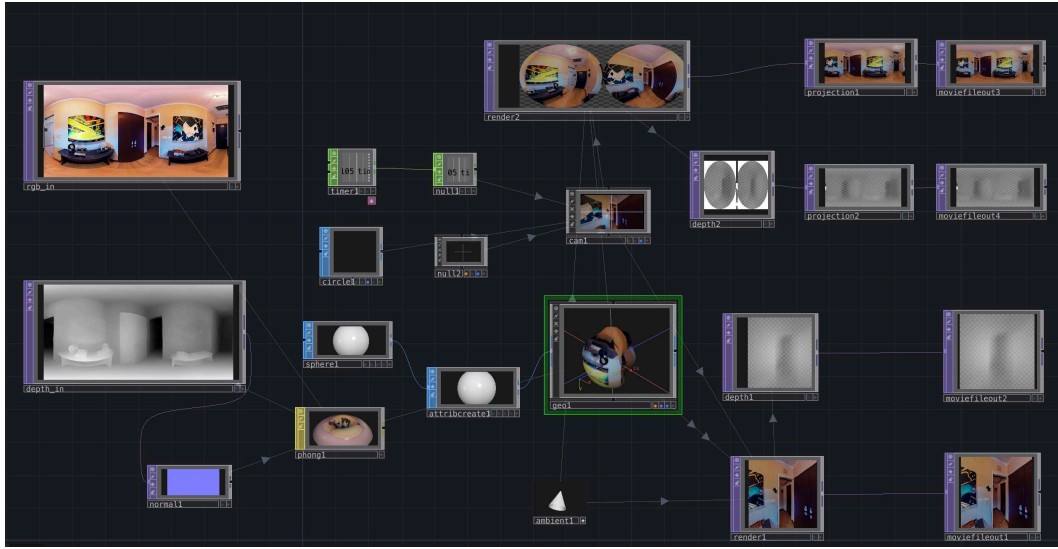

Figure A1: Rendering pipeline of TouchDesigner for LDM3D.

## C    PROMPT LIST USED FOR EVALUATION

During evaluation, each method generates scenes based on 10 indoor scene prompts. Here we list the prompts:

- a bedroom, realistic photo style, 4k
- a dining room, realistic detailed photo, 4k
- a living room, realistic detailed photo, 4k
- a museum exhibition hall displaying sculptures
- a car exhibition center, realistic photo style, 4k
- a study room, realistic detailed photo, 4k
- a table tennis room, realistic detailed photo, 4k
- a washing room, realistic detailed photo, 4k
- a classroom, realistic detailed photo, 4k
- a computer laboratory, realistic detailed photo, 4k

## D    FRÉCHET INCEPTION DISTANCE OVER DISPARITY MAP

As we observe severe visual artifacts exist in rendered disparity map of baseline methods (floating, distortion, blurriness, and discontinuity), we would like to use Fréchet Inception Distance (FID) to assess the image quality. FID is commonly used to evaluate generators trained on real-world images. It utilizes a backbone network that is pretrained on general vision tasks to extract features from each image. By comparing the feature distributions of real dataset images and synthesized fake images, FID quantifies the divergence between these two distributions. Naturally, a question arises regarding the robustness of the backbone network, specifically Inception-v3 in our case, to provide meaningful features that can effectively differentiate between real and fake disparity maps.

To answer this question, we conduct an experiment to verify whether FID exhibits a positive correlation with changes in disparity image quality. To approximate variations in image quality, we test different types of degradatations of the real data, proposed in Heusel et al. (2017): **Gaussian noise** is used to approximate the floating artifacts, **Gaussian blur** is used to approximate blurriness, **Swirl** is used to approximate global distortion, and **Implanted black rectangles** is used to approximate discontinuity and hollows. We use NYU-dep-v2 (Nathan Silberman & Fergus, 2012)

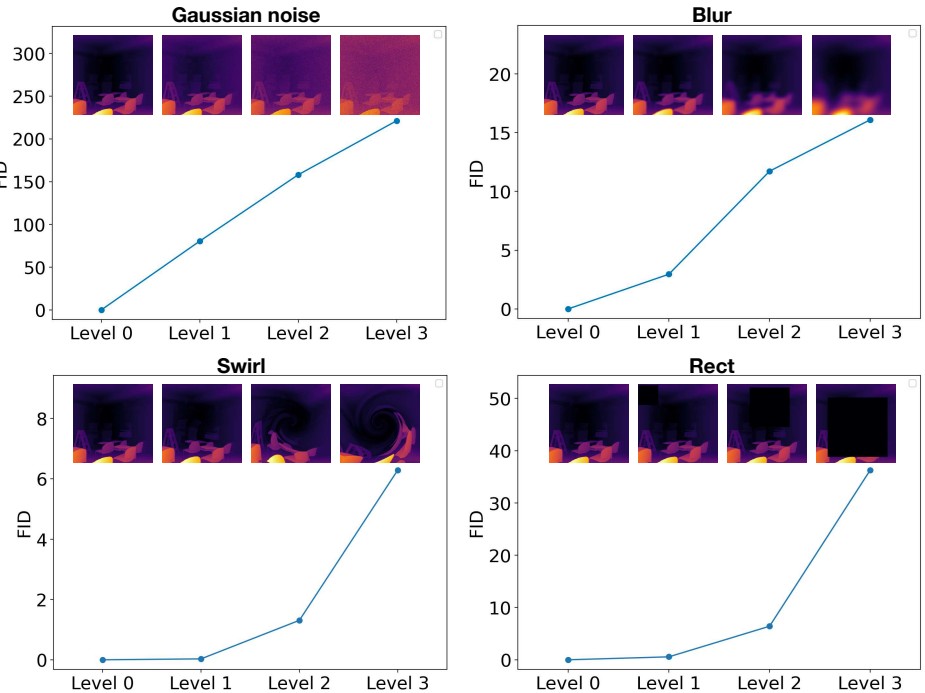

Figure A2: FID is evaluated for **upper left**: Gaussian noise, **upper right**: Gaussian blur, **bottom left**: swirled images, **bottom right**: implanted black rectangles. The disturbance level rises from zero and increases to the highest level. The FID captures the disturbance level very well by monotonically increasing.

as the ground truth dataset. This dataset and the images generated from 10 prompts have significant domain shift globally (e.g different number of objects and different types of scenes). On the other hand local pattern such as edges and surfaces should be pretty similar in these two datasets. To this end we should select the metric that is sensitive to local degradation types, such as **Gaussian blur** and **Gaussian noise**, but largely ignore global transformations such as **Swirl**. We first test the features from different layers of Inception-v3, including 64, 192, 2048 channels.

As shown in Fig. A3, initial feature maps with 64 channels is not sensitive to **Gaussian blur**. On the other hand, global features with 2048 channels exhibit an overly intense response to **Swirl**. To this end we opt to utilize features with 192 channels for calculating the FID in all of our experiments, which adequately captures both local transformation, **Gaussian blur** and **Gaussian noise**, but is almost indifferent for global **Swirl** transformation.

Finally, we shown in Fig. A2, that FID with 192 channels adequately captures different disturbance levels. This justifies our choice of 192 features FID as an evaluation metric.

## E    ARTISTIC SCENES SYNTHESIS

SceneWiz3D can generalize to artistic scenes with diverse styles. We show a gallery of bedrooms with varying styles in Fig. A4.

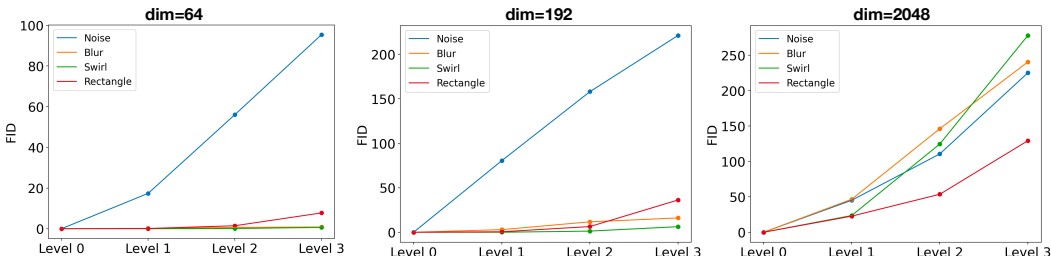

Figure A3: The FID score is calculated for various feature levels, including 64, 192, and 2048 channels. Among these, the mid-level feature with 192 channels exhibits a favorable balance between different types of noise.

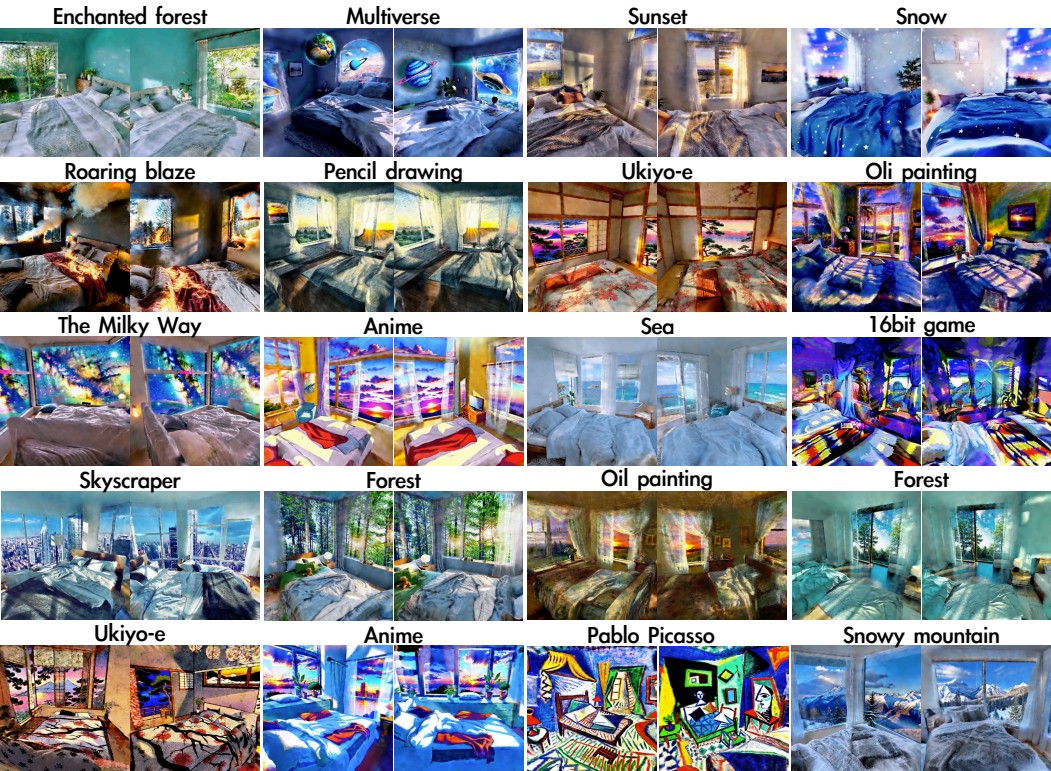

Figure A4: Diverse artistic bedroom scenes generated by `SceneWiz3D`. `SceneWiz3D` can successfully generalize to a wide range of scene characteristics, encompassing various artistic styles and scenic views. With our proposed hybrid representation, we can simultaneously maintain multi-view consistency for foreground objects and synthesize background with varying structure (from skyscraper, sunset, the Milky Way, to the smoke spreading from the burning house).

