# OpenReview forum: "Towards Text-guided 3D Scene Composition"
_ICLR.cc/2024/Conference — ICLR 2024 Conference Withdrawn Submission_

### Official Review · Reviewer_mrx5 · 2023-10-21

**Soundness:** 2 fair
**Presentation:** 1 poor
**Contribution:** 3 good
**Rating:** 3
**Confidence:** 3

**Summary:**

The paper introduces a pipeline (called SceneWiz3D) for text-to-3D scene generation. The proposed method utilizes a hybrid approach of an explicit representation for objects of interest and an implicit representation for scenes. For objects a tetrahedra-based representation is used, while scenes (the background) are represented as a NeRF. The paper discusses a number of experiments and an ablation study. Results are shown as figures and as accompanying supplementary material.

**Strengths:**

- Learning to generate 3D representations from text prompts is an interesting an open research problem.
- Combining implicit and explicit representations for learning 3D scenes appears interesting and novel.
- The paper discusses a number of experiments and ablation studies.

**Weaknesses:**

- The paper lacks a better exposition: several sections are either too verbose (e.g. the introduction) or fail to describe the method with appropriate detail (e.g. Section 3.2 and 3.3.1).
- The paper seems to over-claim the contribution - the claim that objects can be edited is not backed up by meaningful results.
- The shown results are not convincing. Compared to existing approaches, the improvements only seems marginal.
- The advantages of a hybrid (implicit and explicit representation) have not been shown in a convincing manner.

**Questions:**

- The introduction can be more concise and to the point: e.g. the explanation of PSO seems unnecessary and does not add anything.

- While the paper states that scenes can be edited, it is not clear to what extend this is actually possible. No results are shown in the paper that showcase that scenes can be edited. The provided supplementary shows that specific objects can be translated -- these results are not convincing as there are several visual artifacts for the objects in motion. Also it is quite distracting to show camera motion and object motion at the same time. Are the shown object translations the only transformation that can be performed? What about scale or rotation? To further justify the hybrid representation I would suggest to more carefully validate the editability of objects.

- Section 3 appears convoluted and would benefit from a major revision. Figure 2 enables to understand how the components of the model are put together, but the accompanying text is difficult to understand.

- Section 3.2: "While explicit representations, like meshes, are costly to maintain and update when representing a scene with multiple objects." This statement is not clear. Several computer graphics applications relying on explicit representations (e.g. games). Why would a mesh be costly to maintain? Please be more specific.

- When I am not mistaken the symbols for Eq. (1) and (2) are not defined anywhere.

- Section 3.2: What is meant by 'hashing-based encoding'?

- Section 3.2.1: "To optimize the parameters of a scene". What are the parameters?

- Section 3.2.1: "VSD guidance" does not seem to be defined somewhere.

- Section 4.4: "does not excel in fine-grained manipulation tasks". Why should the model perform well on manipulation tasks? What is meant by manipulation here?

**Details Of Ethics Concerns:**

No ethical concerns regarding this submission.

---

### Official Review · Reviewer_RBCq · 2023-10-27

**Soundness:** 3 good
**Presentation:** 3 good
**Contribution:** 3 good
**Rating:** 6
**Confidence:** 3

**Summary:**

This paper proposes to solve 3D scene generation in a more composition-aware manner. Instead of treating foreground and background in the same model, this paper utilises a hybrid representation, which represents objects using DMTet and represents background using NeRF. Representing individual objects explicitly allows users to manipulate objects (including adding, deleting and moving) easily, while proposing new challenges for optimising scene layouts.  To solve it, this paper proposes to use Particle Swarm Optimisation (PSO), which leads to better optimisation results. The whole scene is optimised by SDS/VDS  and CLIP similarity loss.

**Strengths:**

1. This paper proposes to represent each object individually, which generate the whole scene in a more controllable and composition-aware manner. This is a quite promising direction and this paper gives fairly good results.
2. This paper introduces several interesting components into the whole process, like hybrid representation (DMTet + NeRF), and PSO optimisation.

**Weaknesses:**

1. It is hard to tell the contributions of new components from this paper. Although representing objects using DMTet, PSO optimisation, L_pano are very interesting, it is hard to tell whether they actually contribute to results. The whole idea of this paper is quite similar to Dreamfusion and ProlificDreamer, which would be two important baselines. But it is hard to say results of this paper are clearly better than DreamFusion or ProlificDreamer either from quantitive results (Table 1) or qualitative results (images and videos).
2. Related to point 1, table 1 is a little bit confusing.  I assume method with only SDS (or VSD)  in the right table should have similar results as DreamFusion (or ProlificDreamer) in the left table. Then SDS + PSO gives worse results than DreamFusion (SDS only) for CLIP-AP and align, and  VSD + PSO is also worse than ProlificDreamer. Could this be explained somehow?
3. More intermediate results could be helpful to demonstrate the effectiveness of proposed components. More specifically, what is the initial scene configuration? How does the scene configuration changes along with the optimisation? Would the mesh/textures of objects change during optimisation? These intermediate results could be helpful.

**Questions:**

My main concern is: it is hard to tell whether the new components are really helpful or not. More elaboration would be very helpful.

---

### Official Review · Reviewer_QuQy · 2023-10-29

**Soundness:** 3 good
**Presentation:** 3 good
**Contribution:** 3 good
**Rating:** 6
**Confidence:** 4

**Summary:**

This paper presents an novel approach for text-guided 3D scene generation, which decouples the generation of objects and background. For object generation, the method utilizes a readily available text-to-3D model. In terms of background generation, the method employs the NeRF representation and VSD/SDS-based optimization using 2D diffusion networks. In addition, the paper harnesses the capabilities of the CLIP model to optimize object configurations and introduces an innovative optimization technique termed 'particle swarm optimization'.

**Strengths:**

1. The proposed method is clear and well-motivated.

2. The paper is well-written and easy to follow in general.

3. The paper provides extensive evaluations and demonstrates the superiority of the proposed method.

4. The paper also provides many insightful ablation studies to analyze each component.

**Weaknesses:**

1. The method integrates several components, including LLM, CLIP, two diffusion models, and DMTet. The robustness of the proposed method remains ambiguous, and it's uncertain how the limitations of each module impact the overall performance.

2. The utility of the generated 3D scene in downstream applications is not clear. The authors should offer concrete examples demonstrating the practical application of these generated scenes.

3. The proposed "Particle Swarm Optimization" appears to be an empirical method lacking substantial theoretical analysis and assurance. I'm also curious about how it compares with existing optimization strategies, such as simulated annealing.

4. The method employs SDFusion for object generation. This method, however, may not support the generation of open-world objects as it relies on a limited 3D dataset. Opting for alternative text-to-3D methods could notably increase the runtime.

5. I'm skeptical about the robustness of the CLIP score in distinguishing between various scene configurations. Could you provide more comprehensive explanations or results to affirm its efficacy? Have there been any considerations to use the SDS loss to guide scene configuration optimization?

6. The system's runtime is not stated, and it remains uncertain as to how many objects the method typically produces in one scene.

7. Typographical error: On page 5, it should read "trial".

**Questions:**

It's unclear whether the texture and geometry of the objects are optimized by the SDS and VSD loss. If they are, how do you ensure the 3D consistency of the object texture? Is it more challenging to optimize the texture of the shape through scene renderings? If they are not optimized by the SDS and VSD loss, how do you generate the texture for the objects?

---

### Official Review · Reviewer_YLgn · 2023-10-29

**Soundness:** 3 good
**Presentation:** 2 fair
**Contribution:** 3 good
**Rating:** 5
**Confidence:** 4

**Summary:**

The paper introduces a SceneWiz3D method to generate 3D scenes from text prompts. It models 3D objects as explicit representations (DMTet), while the 3D background as an implicit representation (NeRF). The 3D object poses are optimized by Particle Swarm Optimization (PSO) method, and the scene models (eg., the background) are optimized by minimizing the errors between rendered perspective RGB images and panoramic RGBD images. Experiments demonstrate that the proposed method achieves high-quality results.

**Strengths:**

1. It is very interesting to adopt hybrid shape representations for foreground objects and the background layout separately. This is different from existing 3D generation works which usually use a single representation to model the entire 3D scene.

2. Separately optimizing object models by PSO and the background by other three losses is reasonable and effective.

3. The final performance of 3D scene generation looks great both quantitively and qualitatively.

**Weaknesses:**

1. From my understanding, the key technique contributions lie in Section 3.3. It's suggested to remove Section 3.1 "Preliminaries" and condense Section 3.2. The space should be saved for the subsequent Section 3.3.

2. In Section 3.3.1, the part "Particle swarm optimization" is too brief. There are many points unclear to me. For example, what does the optimization target a_i mean? What are the physically meaning of hyperparameters c_1/r_1/c_2/r_2 in the context of this paper? Why cannot we combine them as fewer hyperparameters? What are the meanings of pbest_i/gbest_i and how to determine them? Since PSO is not gradient based optimization, how to guarantee/prove the searched values are "an optimal solution" as claimed by the paper? Is the searching strategy stable and what are the stopping criteria?

3. In Section 3.3.2, the part "Score distillation sampling in RGBD panoramic view" is too brief. It's suggested to write down a piece of step-by-step algorithm. The paper states that "a version of LDM3D is finetuned over panoramic ...". How do finetune? What are the datasets?

4. In Section 3.3.2, the part "Depth regularization" uses pretrained MiDaS model to predict depth for RGB image. How about the accuracy and how to make sure the predicted depth scales are consistent over multi-views?

5. In experiments, the paper states that DreamFusion and ProlificDreamer are inferior due to the lack of multiview supervision. Is it because the paper does not include the same level of supervision signals to adapt the baselines? If this is case, it is suggested to modify the baselines given the same amount of supervision signals. Otherwise, the comparison is unfair.

6. In evaluation, the paper states that the proposed method can generate various scene types. However, there are no quantitative results to support it, although multiple qualitative results are provided. Similarly, it's advised to quantitively evaluate the manipulation results.

Overall, the core technique parts about optimizing object and scene representations are lack of clarity. More quantitative results should be given to support  the claims.

**Questions:**

Discussed above.